# Characteristics of Soil Parameters of Agricultural Land Use Types, Their Location and Development Forecast

**Jozef Vilček** [1,2], **Štefan Koco** [1,2,\*], **Eva Litavcová** [3] **and Stanislav Torma** [1]

[1] National Agricultural and Food Centre, Soil Science and Conservation Research Institute, 08001 Prešov, Slovakia; jozef.vilcek@nppc.sk (J.V.); stanislav.torma@nppc.sk (S.T.)

[2] Department of Geography and Applied Geoinformatics, University of Prešov, 08001 Prešov, Slovakia

[3] Department of Mathematical Methods and Managerial Informatics, University of Prešov, 08001 Prešov, Slovakia; eva.litavcova@unipo.sk

\* Correspondence: stefan.koco@unipo.sk

**Abstract:** In this paper we point out the basic soil parameters characterizing current arable land, permanent grassland, vineyards, and orchards in Slovakia. While the area of permanent land use types is more or less stable, there is a noticeable decrease in the area of arable land. In Slovakia, arable land is located mainly on the plain. The value of its production potential is 67 points (the highest quality soil has 100 points). Permanent grassland is found at higher altitudes on slopes, with a higher gravel content, and the value of their production potential is 35 points. Vineyards are predominantly located in the warm regions of southern Slovakia on the middle slopes. These soils are generally loamy, without significant gravel content, and the value of their production potential is 59 points. Most orchards are located on the plains. The soils are predominantly loamy and deep, without significant gravel content, and the value of their production potential is 63 points. Characteristics of agricultural land use types were determined using vector databases of soil parameters obtained from Soil Science and Conservation Research Institute information systems and a current vector layer for identification of agriculturally used soils, the Land Parcel Identification System, using geographic information systems. Moreover, our analysis tries to determine what developments can be expected in the use of four agricultural land use types. The modeling assumptions concern the future performance of these variables using exponential smoothing and Box–Jenkins methodology.

**Keywords:** agricultural land use type; arable land; permanent grasslands; vineyards; orchards; geographic information system (GIS)

## 1. Introduction

The agricultural landscape in Central Europe is an important landscape element. Its condition and use significantly affects not only the economy of agriculture but also the ecological stability of the environment. The human population, its constant growth, and the related exploitation of natural resources play important roles in this process [1]. In order to ensure food sufficiency, areas of agricultural soil are expanded at the expense of soil originally intended for other uses (e.g., ecological) [2]. According to Foley et. al. [3], up to 40% of the world's soil (which was originally forest, savanna, and grassland) is currently used for agriculture. The ecosystems of the Earth created by long-term natural development under the influence of humans have changed, and their stability and natural balance have been disturbed. The ability to provide ecosystem as well as production and trophic services is decreasing. Land use change can also significantly modify regional and global climates [4,5]. The expansion of soil areas used by farmers is largely responsible for this [6–8].

Awareness of the consequences of human activities, the search for opportunities to reduce environmental disruption, thorough registration and analysis of the geographic distribution and changes in agricultural land use, the creation of databases on soil parameters, and the use of remote sensing methods and other modern information technologies can significantly contribute to rational use of our planet (e.g., [9–12]).

The goal of this paper is to analyze and present landscape use by individual agricultural land use types (arable land, permanent grassland, vineyards, and orchards) in the Slovak Republic territory. We want to also point out the main physical, geographic, qualitative, production, and non-production parameters of these areas and their suitability for such use. For these purposes, geographic information system tools of ArcGIS (ESRI, Redlands, CA, USA) were used for quantification and spatial representation [13]. Part of the goal of this research is to find out what development can be expected in the future. For this purpose, mathematical models for prediction of possible development (share and area) of analyzed soil use types in Slovakia were used.

We believe that achieving this goal (knowledge of soil parameters and potential) will contribute to more effective use and increased ecological stability of the Slovak landscape. The model of expected and projected forecast of individual soil type development can be used in creating state agriculture policy, and in planning and estimating agricultural activities. Nowadays in Slovakia, agricultural land occupies up to 48.6% of the territory. Among other factors, soil-ecological parameters significantly contribute to its use. Soil properties are a decisive factor in terms of real, registered, and statistical soil use. The Statistical Yearbook on Soil Fund of Slovakia [14], based on the cadastre of the Slovak Republic, states that as of 1 January 2018 there were 1,408,660 ha of arable land, 510 ha of hop gardens, 26,258 ha of vineyards, 76,111 ha of gardens, 16 558 ha of orchards, and 853,757 ha of permanent grassland (Figure 1).

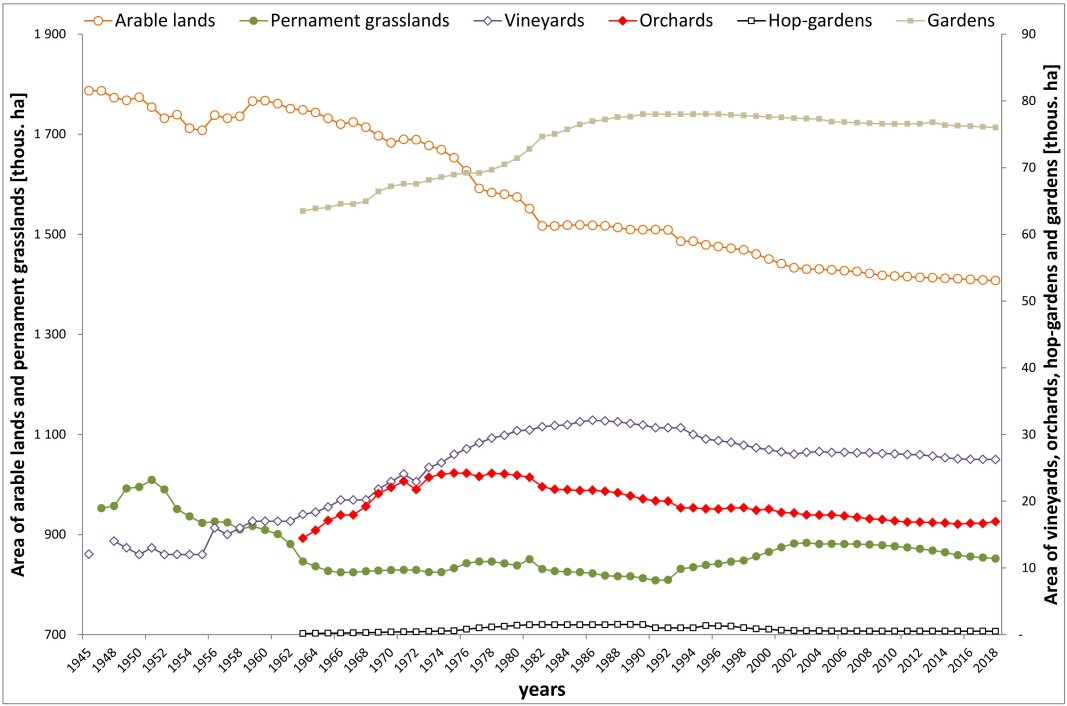

**Figure 1.** Area evolution of agricultural land use types in Slovakia (1945–2018).

However, the real state of soil use is different from the registered soil use. The reality is better documented by the Land Parcel Identification System (LPIS) registry, which is a component of the Integrated Administration and Control System (IACS) (Figure 2).

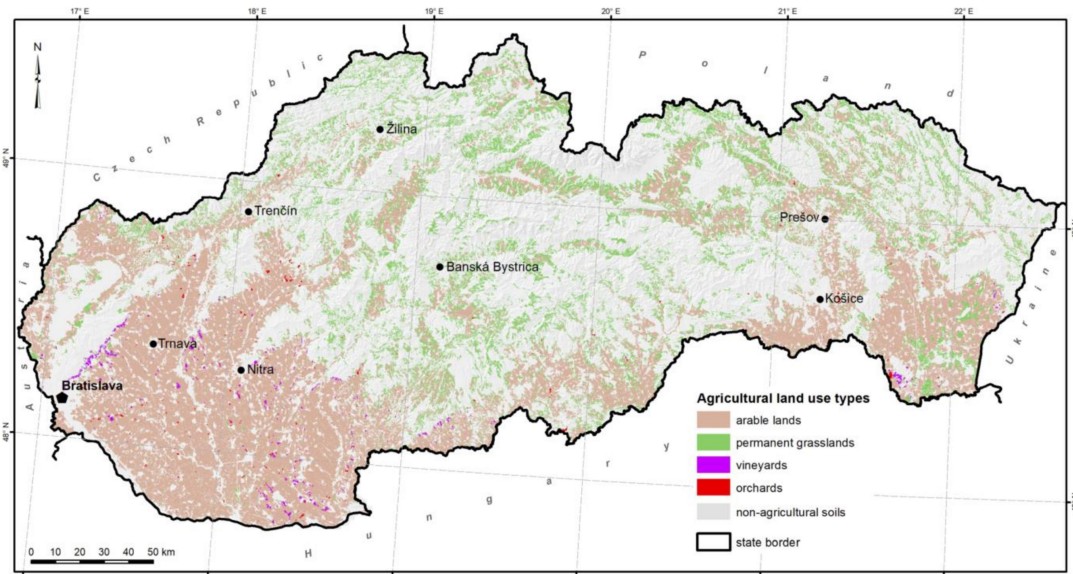

**Figure 2.** Spatial identification of agricultural land use types in Slovakia (2018).

Currently, there are 0.44 ha of agricultural and 0.26 ha of arable land per capita in Slovakia. In the world, these ratios are 0.80 and 0.27 ha [15]. The most arable land per capita is in Australia (1.90 ha), Kazakhstan (1.65 ha), Canada (1.21 ha), Argentina (0.90 ha), and the Russian Federation (0.85 ha) [16].

## 2. Materials and Methods

In the Slovak Republic, the LPIS soil registry provides objective information about the use of the agrarian landscape. It is based on parts of soil blocks that represent agricultural areas with stable boundaries. LPIS represents the vector boundaries of the agrarian landscape and carries information about the size and type of land use of registered areas. The LPIS soil registry is the starting point for information to identify and quantify the analyzed agricultural land use types. The obtained data and LPIS database were processed into the required form for the purpose of our research (based on orthophotomaps) and then subjected to existing soil parameter databases (BPEJ database [17]; soil portal VÚPOP: http://www.podnemapy.sk/default.aspx) in the ArcGIS environment, as shown in Figure 3.

From the soil parameter databases, data relevant to the physical, geographic, and qualitative characteristics of dominant land types in the agrarian country of Slovakia were used.

In concrete terms, the information included the following:

1. A categorization of Slovakia according to agroclimatic regions [17] (Table 1);
2. Identification of extreme values of altitudes of individual agricultural land use types (digital elevation model of the Slovak Republic, DMR3.5) [14];
3. A categorization of soil units, slope, depth, gravel content, and texture of agricultural soils [17];
4. Productivity categorization and point values of production potential of agricultural soils (Table 2) [18].
5. Differentiation of environmental potential indices [21]:

   - very low environmental potential: 0–20 points
   - low environmental potential: 21–40 points
   - medium environmental potential: 41–60 points
   - high environmental potential of soils: 61–80 points
   - very high environmental potential: 81–100 points

6. Differentiation of integrated quality indices of agricultural soils [13]:

- very-high-quality soils: index 1
- high-quality soils: index 2
- medium-quality soils: index 3
- low-quality soils: index 4
- very-low-quality soils: index 5

7. Differentiation of potential of phytomass dry matter production [22]:

- very low phytomass production: less than 8 t ha$^{-1}$
- low phytomass production: 8–10 t ha$^{-1}$
- medium phytomass production: 10–12 t ha$^{-1}$
- high phytomass production: 12–14 t ha$^{-1}$
- very high phytomass production: more than 14 t ha$^{-1}$

8. Bioenergy potential of agricultural soils in Slovakia [15,23,24]:

- very low production: less than l4l GJ ha$^{-1}$
- low production: 141–176 GJ ha$^{-1}$
- medium production: 176–212 GJ ha$^{-1}$
- high production: 212–247 GJ ha$^{-1}$
- very high production: more than 247 GJ ha$^{-1}$

**Table 1.** Chosen parameters of soil and climatic regions in framework of Slovakia.

| Code | Characteristics | TS > 10 °C | CMI (mm) | Tveget °C |
|---|---|---|---|---|
| 00 | very warm, very dry, flat | >3000 | 200 | 16–17 |
| 01 | warm, very dry, flat | 3000–2800 | 200–150 | 15–17 |
| 02 | sufficiently warm, dry, hilly | 2800–2500 | 150–100 | 15–16 |
| 03 | warm, very dry, flat, continental | 3160–2800 | 200–150 | 15–17 |
| 04 | warm, very dry, basin-like, continental | 3030–2800 | 200–100 | 15–16 |
| 05 | relatively warm, dry, basin-like, continental | 2800–2500 | 150–100 | 14–15 |
| 06 | relatively warm, moderately dry, highland-like continental | 2800–2500 | 100–50 | 14–15 |
| 07 | moderately warm, moderately moist | 2500–2200 | 100–0 | 13–15 |
| 08 | moderately cold, moderately moist | 2200–2000 | 100–0 | 12–14 |
| 09 | cold, moist | 2000–1800 | 60–50 | 12–13 |
| 10 | very cold, moist | <1800 | <50 | 10–11 |

Notes: TS > 10 °C: sum of average daily air temperatures more than 10 °C; CMI (mm): climatic moisture indicator (difference of potential evaporation and precipitation) according to Džatko and Sobocká, Škvarenina et al., and Tomlain [17,19,20]; Tveget °C: average air temperature during vegetation period.

**Table 2.** Productivity categories of agriculture soils and their point values in Slovakia [18].

| Code | Characteristics | Average Point Value (on a 100-Point Scale) |
|---|---|---|
| O1 | Most productive arable soils | 90.7 |
| O2 | Highly productive arable soils | 84.6 |
| O3 | Very productive arable soils | 71.9 |
| O4 | Productive arable soils | 58.2 |
| O5 | Medium productive arable soils | 48.1 |
| O6 | Less productive arable soils | 38.6 |
| O7 | Low productive arable soils | 28.7 |
| OT1 | Productive arable soils and very productive grassland | 41.3 |
| OT2 | Medium productive arable soils and medium productive grassland | 33.0 |
| OT3 | Low productive arable soils and less productive grassland | 25.4 |
| T1 | Very productive grassland | 24.3 |
| T2 | Medium productive grassland | 22.0 |
| T3 | Less productive grassland | 14.7 |

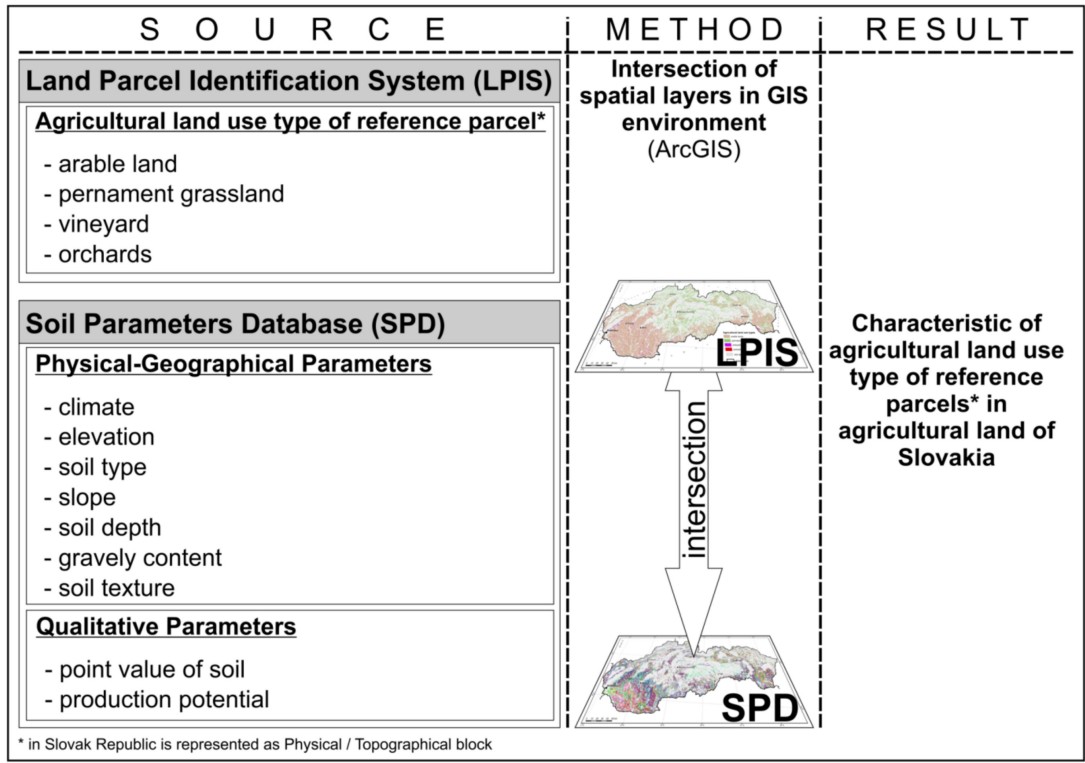

**Figure 3.** Scheme of methodology.

For the purposes of modeling the forecast of areas of agricultural land use in the coming years (2021–2024), we used Brown [25] and Holt's [26] exponential smoothing methods. Exponential smoothing to forecast time series is the most commonly used method. Brown's double exponential smoothing uses two smoothed series that are centered at different points in time. The formula can be described as below [27]. $S'$ denotes the single-smoothed series obtained by applying simple exponential smoothing to series $Y$ and $S''$ denotes the double-smoothed series obtained by applying simple exponential smoothing using $\alpha$, the same constant smoothing factor to series $S'$:

$$S'_t = \alpha Y_t + (1 - \alpha)S'_{t-1} , \tag{1}$$

$$S''_t = \alpha S'_t + (1 - \alpha)S''_{t-1}. \tag{2}$$

The forecast for $Y_{t+k}$ for any $k > 1$ is given by

$$\hat{Y}_{t+k} = L_t + kT_t , \tag{3}$$

where $L_t$ is estimated level and $T_t$ is estimated trend at period $t$, and they are given by

$$L_t = 2S'_t + S''_t , \tag{4}$$

$$T_t = \frac{\alpha}{1 - \alpha}(S'_t - S''_t). \tag{5}$$

For the purpose of model-fitting (i.e., calculating forecasts, residuals, and residual statistics over the estimation period), the model can be started up by setting $S'_0 = S''_0 = Y_1$ (i.e., set both smoothed series equal to the observed value at $t = 1$).

Another approach, the Box–Jenkins methodology, uses autoregressive integrated moving averages (ARIMA) and seasonal ARIMA (SARIMA), if a seasonal component is present. According to previous

research [28–30], a simple equation to define the autoregressive moving average (ARMA) model for a stationary time series is as follows:

$$Y_t = \varphi_1 Y_{t-1} + \varphi_2 Y_{t-2} + \ldots + \varphi_p Y_{t-p} + \varepsilon_t - \theta_1 \varepsilon_{t-1} - \theta_2 \varepsilon_{t-2} - \ldots - \theta_q \varepsilon_{t-q}. \tag{6}$$

The first term in the ARMA model represents an autoregressive AR($p$) term of the order $p$ having the form of

$$Y_t = \varphi_1 Y_{t-1} + \varphi_2 Y_{t-2} + \ldots + \varphi_p Y_{t-p} + \varepsilon_t. \tag{7}$$

This AR($p$) term refers to the current time series value $Y_t$ as a function of past time series values $Y_{t-1}, Y_{t-2}, \ldots, Y_{t-p}$. The variables $\varphi_1, \varphi_2, \ldots, \varphi_p$ are autoregressive coefficients that relate $Y_t$ to $Y_{t-1}$, $Y_{t-2}, \ldots, Y_{t-p}$. The moving average MA($q$) term of the model is represented as

$$Y_t = \varepsilon_t - \theta_1 \varepsilon_{t-1} - \theta_2 \varepsilon_{t-2} - \ldots - \theta_q \varepsilon_{t-q}, \tag{8}$$

where $\varepsilon_{t-1}, \varepsilon_{t-2}, \ldots, \varepsilon_{t-q}$ are past random shocks or independent white noise sequences with mean = 0 and constant variance = $\sigma^2$, and $\theta_1, \theta_2, \ldots, \theta_q$ are the moving average coefficients relating $Y_t$ to $\varepsilon_{t-1}, \varepsilon_{t-2}, \ldots, \varepsilon_{t-q}$. Stationarity of the time series is achieved by its differentiation. Then the model notation is ARIMA($p,d,q$), where the AR and MA specifications are combined with the integration (differencing) term. The letters $p$, $d$, and $q$ indicate orders of autoregression, differencing, and moving average, respectively. The model is mathematically given as

$$(1 - B)^d Y_t = \frac{\theta(B)}{\varphi(B)} \varepsilon_t, \tag{9}$$

where $t$ denotes the time indices, $B$ is the backshift operator (i.e., $BY_t = Y_{t-1}$), and $\varphi(B)$ and $\theta(B)$ are the autoregressive and moving average operators, respectively, and can be written as

$$\varphi(B) = 1 - \varphi_1 B^1 - \varphi_2 B^2 - \ldots - \varphi_p B^p \tag{10}$$

$$\theta(B) = 1 - \theta_1 B^1 - \theta_2 B^2 - \ldots - \theta_q B^q. \tag{11}$$

To determine the validity of our models, goodness-of-fit statistics were used. We used the criterion of the mean absolute percentage error (MAPE). The MAPE is a measure of how much a dependent series varies from its model-predicted level [31]. It is independent of the units used and can therefore be used to compare series with different units, as given by

$$MAPE = \sum_{t=1}^{n} \left| \frac{(Y_t - F_t)/Y_t}{n} \right| \times 100. \tag{12}$$

In this equation, $Y_t$ is the actual data for period $t$, $F_t$ denotes the forecast for period $t$, and $n$ is the number of observations. MAPE $\leq$ 10% means a highly accurate forecast. Another criterion we used was stationary $R^2$, given by IBM SPSS Statistic [31]

$$R_S^2 = 1 - \frac{\sum_t (Y_t - F_t)^2}{\sum_t \left( \Delta Y_t - \overline{\Delta Y} \right)^2}. \tag{13}$$

Stationary $R^2$ is a measure that compares the stationary part of the model to a simple mean model. This measure is preferable to ordinary $R^2$ when there is a trend or seasonal pattern. Stationary $R^2$ can be negative, with a range of negative infinity to 1. Negative values mean that the model under consideration is worse than the baseline model; positive values mean that the model is better.

Parsimonious models are simple models with great explanatory predictive power. They explain data with a minimum number of parameters or predictor variables. To compare models in order to

find a parsimonious model, Akaike information criterion (AIC) [32] and normalized Bayes information criterion (BIC) [33] were used. Additional tests were used to determine if the assumptions were met. To test the normality of residuals, Shapiro–Wilk test (SW) was used [34]. To test the null hypothesis that there was no autocorrelation in residuals, Ljung–Box Q statistics (LB-Q) [35] were used.

The model variables for arable land and orchards in Slovakia were used, corresponding to yearly unadjusted data of these variables from 1996 to 2018, permanent grasslands from 1997 to 2018, and vineyards from 1973 to 2018. Examining these chosen time series made it possible to find the resulting parsimonious forecasting models in which all necessary assumptions were met.

## 3. Results

### 3.1. Arable Land

Arable land occupies 28.7% of the total area of Slovakia and 59.0% of agricultural soil. The highest proportion of arable land (64.7%) is in the lowland areas (50.2% in the Danube lowland), and the lowest in mountain and piedmont regions (16.4%). It is located at altitudes of 95 m a.s.l. up to 1170 m a.s.l., and the average altitude of arable land is 220 m a.s.l. According to the agroclimatic regionalization of Slovakia [17], more than half of the arable land belongs to climate regions 00 (30.2%) and 01 (20.4%) (Table 3). The least arable land is located in climate region 09 (1.9%).

**Table 3.** Share of agroclimatic regions (%) in agricultural land use types in Slovakia.

| Agroclimatic Region | Agricultural Land Use Type | | | |
| --- | --- | --- | --- | --- |
| | Arable Land | Permanent Grassland | Vineyards | Orchards |
| 00 | 30.2 | 0.6 | 26.1 | 28.3 |
| 01 | 20.4 | 2.2 | 28.3 | 14.8 |
| 02 | 8.2 | 3.2 | 17.5 | 17.6 |
| 03 | 9.5 | 4.6 | 7.8 | 6.5 |
| 04 | 7.2 | 2.6 | 4.3 | 2.5 |
| 05 | 6.2 | 5.6 | 2.4 | 9.1 |
| 06 | 3.0 | 6.7 | 3.2 | 3.8 |
| 07 | 6.6 | 19.2 | 0.4 | 12.5 |
| 08 | 3.9 | 15.8 | – | 3.0 |
| 09 | 1.9 | 12.7 | – | 1.5 |
| 10 | 2.8 | 26.8 | – | 0.5 |

Almost 20% of arable land is represented by Chernozems. More than 96% of them are plowed, and Fluvisols, Cutanic Luvisols, and Cambisols are somewhat smaller. Soils such as Podzol, Leptosol, and Solonetz are hardly used as arable land. Arable land is predominant on the plains (64.5%) and slight slopes (3–7°; 22.9%). There are deep soils (more than 0.6 m; 85%), and only rarely medium deep (9%) or shallow (6%). Up to 77.8% of arable land belongs to the category of soils without gravel content, but there are also strongly gravelly soils (6%). According to soil texture, medium-heavy soils dominate, namely sandy (64.5%) and sandy-loam (7.6%). Arable land can also be found on light soils (sandy and loam), sandy, heavy soils (clayey-loam), and very heavy soils (clayey and clay).

A proportion of up to 91.2% of soils actually used as arable land is in accordance with the typological production categorization of soils in Slovakia (Figure 4) [18] for this kind of agricultural land use. About 9% of the soils would be more suitable for use as permanent grassland due to the slope (2% of arable land is on slopes over 12°). The average value of arable land production potential in Slovakia on a 100-point scale [18] is 67 points. Arable land has the same average point value, even in terms of its environmental potential [21]. On average, 11 tons of dry matter are produced per hectare of arable land, which is 216 GJ in energy. According to the evaluation of the soil quality index in Slovakia [13], this value is 2.6 for arable land, which is a category of high- to medium-quality soils.

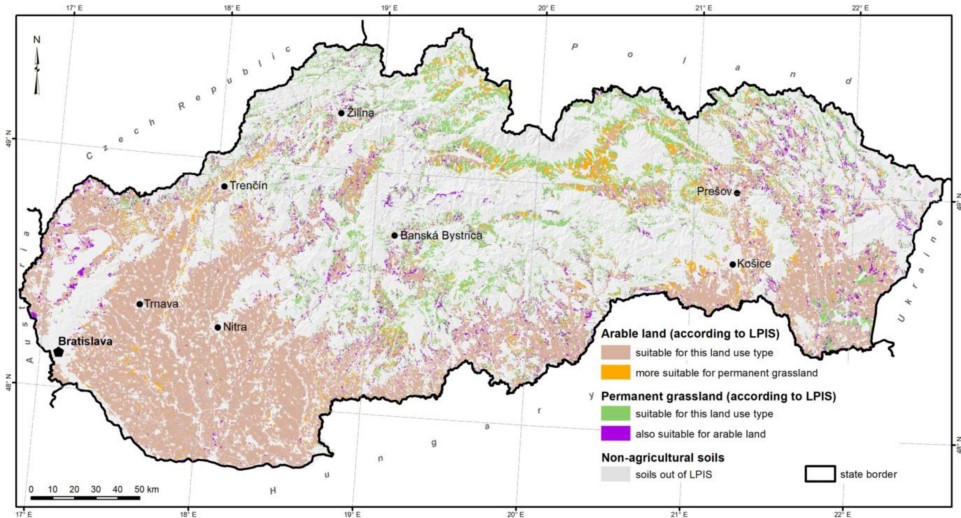

**Figure 4.** Soil suitability for use as arable land and permanent grassland.

To forecast the amount of arable land for the next four years, the ARIMA model was used. Taking into account assumptions and data fitting, we chose the most parsimonious model to test the assumption of no autocorrelation in residuals. The best resulting model was found to be ARIMA(1,1,0), used on naturally logged past data in the interval 1996–2018 with parameter $\varphi_1 = 0.863$ and significant at $p < 0.001$. The model is very good in terms of assumptions too: $p(\text{LB-Q}) = 0.830$, $p(\text{SW}) = 0.150$, MAPE = 0.098, stationary $R^2 = 0.499$. Forecast values for four years are as follows:

- 1,405,654 ha for 2021
- 1,405,155 ha for 2022
- 1,404,732 ha for 2023
- 1,404,376 ha for 2024

The resulting model for the variable arable lands can be seen in Figure 5.

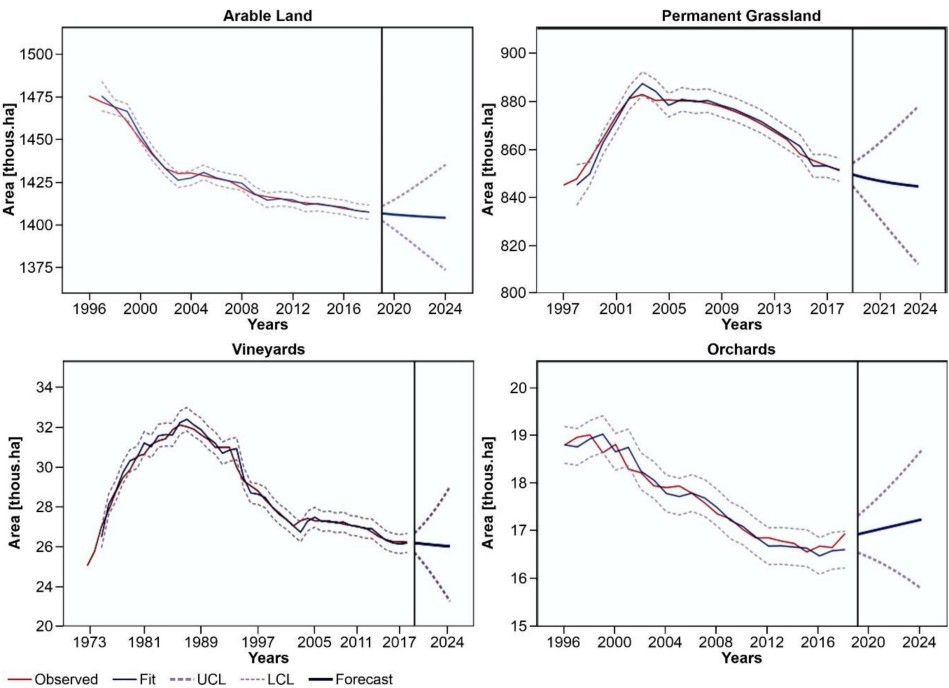

**Figure 5.** Observed, smoothed, and forecast series with 95% confidence bounds of ARIMA or Brown's exponential models of arable land, permanent grassland, vineyards, and orchards at selected time intervals.

### 3.2. Permanent Grassland

Grassland occupies 17.5% of the total area of Slovakia and 35.9% of agricultural soil. The highest proportion of permanent grassland (74.9%) is in the mountain and piedmont regions, and the lowest in the lowlands (8.1%). Grassland is located at altitudes of 94 m a.s.l. up to 1972 m a.s.l, and the average altitude of permanent grassland is 501 m a.s.l. According to the agroclimatic regionalization of Slovakia [17], more than 55% of permanent grassland belongs to cold climate regions 08 (15.8%), 09 (12.7%), and 10 (26.8%). The least permanent grassland is located in very warm climate region 00 (0.6%).

Almost 66.4% of permanent grassland is represented by the Cambisol soil type. Leptosols, Fluvisols, and Planosols have significantly smaller shares. Other soil types are represented only marginally (Table 4). Permanent grassland is predominant on medium (7–12°; 34.9%) and steep (12–17°; 20.3%) slopes. It consists of mostly shallow (less than 0.3 m; 41.2%) and very gravelly (41.1%) soils. According to soil texture, medium-heavy or loamy (53.3%) and sandy-loamy (21.1%) dominate.

A proportion of up to 76.7% of soils actually used as permanent grassland is in accordance with the typological production categorization of soils in Slovakia [18] for this kind of agricultural land use. The average value of permanent grassland production potential on a 100-point scale [18] is 35 points. More than 20% of current permanent grassland is potentially suitable for use as arable land (Figure 4). In terms of the environmental potential assessment of soils [21], the value is 53 points. On average, 5.8 tons of dry matter are produced per hectare of permanent grassland, which is 103 GJ in energy. According to the evaluation of the soil quality index in Slovakia [13], this value is 3.6 for permanent grassland, which is a category of medium- to low-quality soils.

**Table 4.** Share of soil type (%) in agricultural land use types in Slovakia.

| Soil Type | Agricultural Land Use Type | | | |
|---|---|---|---|---|
| | Arable Land | Permanent Grassland | Vineyards | Orchards |
| Fluvisols | 18.0 | 6.5 | 1.7 | 11.0 |
| Chernozem | 19.3 | 0.2 | 20.0 | 20.2 |
| Molic Fluvisol | 11.6 | 1.6 | 0.3 | 4.9 |
| Cutanic Luvisol | 15.0 | 2.2 | 19.7 | 21.0 |
| Regosols | 7.2 | 2.5 | 19.5 | 8.9 |
| Cutanic Albic Luvisol | 2.4 | 1.1 | 1.1 | 3.0 |
| Planosol | 9.3 | 5.5 | 2.7 | 8.2 |
| Cambisol | 13.9 | 66.4 | 24.0 | 21.1 |
| Podzol | – | 0.2 | – | – |
| Rendzic | 1.5 | 8.3 | 1.5 | 1.1 |
| Gleysol | 1.4 | 3.2 | – | 0.1 |
| Solonetz | 0.1 | 0.1 | – | – |
| Leptosols | – | 0.4 | 0.2 | 0.1 |
| Anthrosol | – | – | 9.4 | 0.1 |

To forecast the amount of permanent grassland for the next four years, taking into account assumptions and data fitting, we chose the most parsimonious ARIMA(1,1,0) model on naturally logged past data in the interval 1997–2018 with parameter $\varphi_1 = 0.831$ and significant at $p < 0.001$. The model is very good in terms of assumptions too: $p$(LB-Q) = 0.654, $p$(SW) = 0.144, MAPE = 0.197, stationary $R^2$ = 0.715. Forecast values for four years are as follows:

- 847,390 ha for 2021;
- 846,436 ha for 2022;
- 845,659 ha for 2023;
- 845,029 ha for 2024.

The resulting model of the variable permanent grasslands can be seen in Figure 5.

### 3.3. Vineyards

Vineyards occupy 0.5% of the total area of Slovakia and 1.1% of agricultural soil. The highest proportion of vineyards, 63.5%, is in lowland areas, with 33.5% in mountain and piedmont areas. Almost 40% of the vineyards are situated in the Nitra Hill Land region and almost 7% in the Zemplín Mountain region. Vineyards registered in LPIS are located at altitudes of 96 m a.s.l. up to 460 m a.s.l. According to the agroclimatic regionalization [17], vineyards are predominantly located in the warm regions of southern Slovakia: 26.1% in climate region 00 and 28.3% in region 01. Vineyards are not present in cold regions.

The vineyards are mostly on Cambisol type soil (24%), but also on Chernozem (20%), Cutanic Luvisols (19.7%), and Leptosols (19.5%). Other soil types are represented only marginally. Most vineyards (39.6%) are located on medium slopes (7–12°), and fewer on the plains (28.0%). They are mostly on deep soils (more than 0.6 m; 67.1%) and without a significant gravel content (58.4%). According to soil texture, medium-heavy or loamy dominates (67.7%).

Their average value in Slovakia on a 100-point scale [18] is 59 points. In terms of the environmental potential assessment of soils [21], the value is 65 points. On average, 10 tons of dry matter are produced per hectare of vineyards, which is 194 GJ in energy. According to the evaluation of the soil quality index in Slovakia [13], the value is 2.8 for vineyards, which is a category of high- to medium-quality soils.

To forecast the amount of vineyards for the next four years, taking into account assumptions and data fitting, we chose the most parsimonious ARIMA(0,2,1) model on naturally logged past data for the interval 1973–2018 with parameter $\theta_1 = 0.475$ and significant at $p < 0.01$. The model is good in terms of assumptions too: $p(\text{LB-Q}) = 0.251$, $p(\text{SW}) = 0.051$, MAPE = 0.646, stationary $R^2 = 0.160$. Forecast values for four years are as follows:

- 26,119 ha for 2021;
- 26,084 ha for 2022;
- 26,052 ha for 2023;
- 26,024 ha for 2024.

The resulting model of the variable vineyards can be seen in Figure 5.

### 3.4. Orchards

Orchards occupy 0.3% of the total area of Slovakia and 0.7% of agricultural soil. The highest proportion of orchards, 59.3%, is in the lowland areas, with 30.1% in mountain and piedmont areas. Almost 27% of the orchards are situated in the Nitra Hill Land region. Orchards registered in LPIS are located at altitudes of 97 m a.s.l. up to 818 m a.s.l. According to agroclimatic regionalization [17], orchards are predominantly located in the warm regions of southern Slovakia, with 28.3% in climate region 00.

Orchards in Slovakia occur most on Cambisol type soil (21%), but also on Cutanic Luvisol (21%) and Chernozem (20%). Most orchards (39.6%) are located on the plains (0–3°). They are mostly on deep soils (more than 0.6 m; 78.9%), without a significant gravel content (69.7%). According to soil texture, medium-heavy or loamy dominates (68.7%).

The average value of soils on which orchards are located on a 100-point scale [18] is 63 points. For the environmental potential assessment of soils [21], the value is 65 points. On average, 11 tons of dry matter are produced per hectare of orchards, which is 209 GJ in energy. According to the evaluation of the soil quality index in Slovakia [13], the value is 2.7 for orchards, which is a category of high- to medium-quality soils.

To forecast the number of orchards for the next four years, taking into account assumptions and data fitting, we chose the best Brown exponential smoothing model based on past data in the interval 1996–2018 with level and trend parameter $\alpha = 0.520$ and significant at $p < 0.001$. The model is very

good in terms of assumptions too: $p$(LB-Q) = 0.403, $p$(SW) = 0.138, MAPE = 0.778, stationary $R^2$ = 0.554. Forecast values for four years are as follows:

- 17,059 ha for 2021;
- 17,121 ha for 2022;
- 17,183 ha for 2023;
- 17,245 ha for 2024.

The resulting model of the variable orchards can be seen in Figure 5.

Characteristics and representation of analyzed soil parameters of individual agricultural land use types are given in Tables 3–9.

**Table 5.** Share of soil texture categories (%) in agricultural land use types in Slovakia.

| Soil Texture | Agricultural Land Use Type | | | |
| | Arable Land | Permanent Grassland | Vineyards | Orchards |
|---|---|---|---|---|
| Sandy and loam-sand | 5.3 | 6.2 | 9.5 | 7.4 |
| Sandy-loam | 64.5 | 53.3 | 67.7 | 68.7 |
| Loam | 18.5 | 16.5 | 11.6 | 14.2 |
| Clayey-loam | 4.2 | 2.9 | – | 0.7 |
| Clayey and clay | 7.6 | 21.1 | 11.2 | 8.9 |

**Table 6.** Share of slope degree (%) in agricultural land use types in Slovakia.

| Slope | Agricultural Land Use Type | | | |
| | Arable Land | Permanent Grassland | Vineyards | Orchards |
|---|---|---|---|---|
| 0–3° | 64.5 | 14.4 | 28.0 | 48.0 |
| 3–7° | 22.9 | 18.4 | 39.6 | 24.8 |
| 7–12° | 10.4 | 34.9 | 21.2 | 19.6 |
| 12–17° | 1.9 | 20.3 | 8.0 | 6.4 |
| >17° | 0.3 | 11.9 | 3.2 | 1.2 |

**Table 7.** Share of gravelly soil categories (%) in agricultural land use types in Slovakia.

| Gravelly Soil Categories | Agricultural Land Use Type | | | |
| | Arable Land | Permanent Grassland | Vineyards | Orchards |
|---|---|---|---|---|
| None or sporadic | 77.7 | 17.4 | 58.4 | 69.7 |
| Slight | 11.1 | 25.1 | 17.9 | 15.9 |
| Medium | 5.3 | 16.4 | 7.7 | 6.4 |
| Very gravelly | 6.0 | 41.1 | 15.9 | 7.9 |

**Table 8.** Share of soil depth categories (%) in agricultural land use types in Slovakia.

| Soil Depth Categories | Agricultural Land Use Type | | | |
| | Arable Land | Permanent Grassland | Vineyards | Orchards |
|---|---|---|---|---|
| Deep (>0.6 m) | 85.1 | 31.3 | 67.1 | 78.9 |
| Medium deep (0.3–0.6 m) | 9.0 | 27.6 | 16.9 | 12.8 |
| Shallow (<0.3 m) | 6.0 | 41.2 | 15.9 | 8.3 |

**Table 9.** Share of soil quality categories (%) in agricultural land use types in Slovakia.

| Index of Soil Quality | Characteristic | Agricultural Land Use Type | | | |
| --- | --- | --- | --- | --- | --- |
| | | Arable Land | Permanent Grassland | Vineyards | Orchards |
| 1 | Very-high-quality | 1.7 | 0.0 | 0.6 | 0.8 |
| 2 | High-quality | 48.9 | 3.0 | 39.1 | 44.6 |
| 3 | Medium-high quality | 40.7 | 35.9 | 44.3 | 40.0 |
| 4 | Low-quality | 8.6 | 60.4 | 15.9 | 14.5 |
| 5 | Very-low-quality | 0.1 | 0.7 | 0.1 | 0.1 |
| Average index value | | 2.6 | 3.6 | 2.8 | 2.7 |

## 4. Discussion

There is an abundance of data and statistical comparisons regarding the use of agricultural soils and the landscape around the world. Published and presented data are the result of direct observations and measurements in the field [36,37], deriving their representation and use from existing (historical) maps [38,39], the use of remote sensing methods (global navigation satellite system (GNSS), satellite images, orthophotomaps, etc.) [40–44], modeling and forecasting [45,46], etc. We used a combination of national statistical data with modeling and also applied data from remote sensing analyzed by geographic information system (GIS) tools.

The state and development of the agricultural landscape are statistically monitored by several global databases. Under the Food and Agriculture Organization (FAO) statistics (FAOSTAT) [47] and World Development Indicators (WDI) database definitions, agricultural soils covered 37.4% of the world's land area in 2016. Permanent pastures make up 67.3% of all agricultural soils (25.2% of global land area), arable land (row crops) makes up 29.2% (10.9% of global land area), and permanent crops (e.g., vineyards and orchards) make up 3.4% (2.1% of global land area).

The largest proportions of arable land in the world are located in south Asia (39.4% global land area), Europe (26.8%), southeast Asia (26.3%), and the United States east of the Mississippi River (24.0%). The largest proportions of permanent grassland are located in Argentina, Uruguay, and Chile (33.0%), Pacific countries, China (32.7%), Mexico and Central America (31.4%), the United States west of the Mississippi River (31.4%), and tropical Africa (30.1%). The lowest proportions of arable land are located in Canada, Pacific countries (4.5%), and north of South America (5.4%), and the smallest proportions of permanent grassland are located in southeast Asia (1.7%), Canada (2.3%), and the United States east of the Mississippi River (5.0%) [48]. Globally, the total amount of permanent grassland according to the FAO has been in decline since 1998 [49]. In Europe, agricultural soil covers 44.6% of the territory, where 59.2% is arable land (25.7% of all soil).

Almost all such characteristics indicate either the direct size or the percentage ratio of individual land use types. However, more detailed analysis of the soil conditions at these sites is scarce. In this paper we focused on the analysis and evaluation of parameters and characteristics of agricultural soils in Slovakia, which predominantly decide on how they are used.

A more detailed description of soil parameters as the limits for soil use is absent in statistical reviews (not only in Slovakia, but in the world). Therefore, we tried to point out soil suitability for current use, and not just simple numbers about the status and development of individual agricultural land use types. It turns out that not all soils in Slovakia are used in accordance with their productive potential. For example, some fields on steep slopes, soil with high gravel content, and shallow soils are used as arable soils. Our results correspond with previous research of soil scientists and geographers dealing with landscaping and land use in Slovakia [15,18,22].

Even in the conditions of Slovakia, land assessment is a tool for strategic land use planning. Although there are many simulation models, computer packages for assessing land in spatial planning, economic models of land use [50–52], and their application to local conditions (except for regions for which the model has been calibrated) are largely incorrect and give inaccurate information results.

As early as 1996, Rossiter [53] sought to point out the different levels and practices of model generalization. In this sense he established three categories of soil assessment models: (i) non-spatial models of territory suitability for one area, (ii) spatial models of single-area soil suitability, and (iii) multi-area suitability models for the problem of land allocation. This systematic analysis provides a comprehensive overview of all existing landscape assessment frameworks and places each method in the same framework according to mathematical formulas. When evaluating the use of agricultural soil, we tried to use local information databases and data created and updated by the Slovak pedological community valid for specific soil units and entities.

One objective of this paper was to determine what developments can be expected with the use of four agricultural land use types. To address this objective, we modeled the current and future development of selected variables by using exponential smoothing and Box–Jenkins (ARIMA) modeling. This methodology was used in a number of related studies by Al-Hiyali and Al-Wasity [54] to predict the proportion of agricultural or cultivable land in the whole country, and by Ullah et al. [55] to forecast peach area and production. In a wider context, the methodology can be used to forecast various types of time series, such as by Sena and Nagwani [56] to forecast monthly groundwater levels in India, and Litavcová and Vašaničová [57] to model the development of total nights spent at tourist accommodations in European Union (EU) countries. The relevant literature dealing with statistical forecasting extends to the development of the simple exponential smoothing method by Brown [25] and Holt [26] and some of its extensions by Winters [58]. ARIMA models were developed in the 1970s and have been studied extensively by many researchers. Their theoretical bases were described by Box and Jenkins [28] and later by Box, Jenkins, and Reinsel [29]. Nowadays more widely-used complex models and associated statistical tests have been proposed, for example, by Engle [59], Bolerslev [60], Engle in his Nobel lecture [61], Granger [62], and so on. This is clearly stated, for example, in the works of Wooldridge [63] and Arlt and Arltová [64]. In the interest of the existence and survival of human civilization, it is important to understand how global agricultural soils are changing and to evaluate their implications for a sustainable future (e.g., [3,8]).

The current development and the forecast of the development state of agricultural land use types in Slovakia show the opposite development trend compared to the global data. For example, while in the world there is an increase in arable (agricultural) soils at the expense of forest, in Slovakia the tendency is the opposite, which is confirmed by the models we applied.

Our analysis shows that compared with the world and the rest of Europe, the share of agricultural soils in Slovakia is higher (48.6%). Slovakia is above the European average in its share of arable land (28.7% of the total area) and permanent grassland (17.5%), indicating a significant impact of agriculture on the appearance and use of the landscape. However, over the past 60 years, the gradual decline of agricultural soils and the growth of forest and built-up areas have been evident. According to data from the Statistical Office of Slovak Republic since 1950, agricultural soil has decreased by an average of 6116 ha per year in the country. Currently, about 16 ha of soil are lost every day. On the other hand, the area of forest soils increased by 286,000 ha since 1950 [15].

We consider the results we obtained and presented here as a small contribution to the stabilization and ecologization of the agricultural landscape in Slovakia. The results are presented with the hope that they will be an inspiration, especially for the people responsible for land use (politicians and planners, but also farmers). At the same time, we hope that they also will be an inspiration for further research and forecasting in this field.

## 5. Conclusions

Compared to the world and Europe, the share of agricultural land in Slovakia is higher (arable land and permanent grassland). On the other hand, permanent crops (vineyards, orchards) are less represented.

Typical arable land in Slovakia is located on plains, is deep (over 0.6 m) and medium-heavy (loamy), without gravel content. The average value of arable land production potential is 67 points.

Typical permanent grasslands are located on medium slopes (7–12°), are shallow (less than 0.3 m), medium-heavy (loamy) and strongly gravelly. The average value of their production potential is 35 points.

Vineyards are predominantly located in the warm regions of southern Slovakia on medium slopes (7–12°). They are mostly medium-heavy soils, loamy and deep (over 0.6 m), and without significant gravel content. The average value of their production potential is 59 points.

Most orchards are located on the plains. They are mostly deep soils (over 0.6 m) without significant gravel content. According to the soil texture, medium-heavy loamy soils dominate. The average value of their production potential is 63 points.

The forecast for the expected development of agricultural land use types in Slovakia as analyzed by us confirms continuation of the trend in recent years. It is assumed that in 2024, compared to the current state, there will be a decrease in the area of arable land by almost 2020 ha, the area of permanent grassland by almost 5580 ha, and the acreage of vineyards by approximately 190 ha, and an increase in the area of orchards by 140 ha.

The current use of agricultural land in Slovakia largely reflects production parameters and soil quality. More quality soils are located especially in lowlands and basins and are predominantly used as arable land for food security. It would be good for the government to devote more attention to expanding orchards and vineyards, for instance in the form of systemic subsidies.

We suppose that the knowledge of soil parameters and attributes and the potential of the agricultural landscape will contribute significantly to its rational use. The soil will be able to be used for the purposes for which it is most suitable. Such an approach will contribute to landscape ecologization and economic efficiency.

**Author Contributions:** Conceptualization, J.V.; methodology, J.V., E.L. and S.T.; software, Š.K. and E.L.; validation, J.V., Š.K., E.L. and S.T.; formal analysis, J.V., Š.K. and E.L.; investigation, J.V., E.L., Š.K. and S.T.; resources, J.V., Š.K. and S.T.; data curation, Š.K., S.T. and E.L.; writing—original draft preparation, J.V., Š.K., E.L. and S.T.; writing—review and editing, J.V., Š.K. and E.L.; visualization, J.V. and Š.K.; supervision, J.V.; project administration, J.V. and Š.K.; funding acquisition, J.V. All authors have read and agreed to the published version of the manuscript.

**Funding:** This work was supported by the Slovak Research and Development Agency under Grant No. APVV-15-0406 and Scientific Grant Agency of Ministry of Education of Slovak Republic under Grant No. VEGA 1/0059/19.

**Conflicts of Interest:** The authors declare no conflicts of interest.

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
