# Peer review of "Characteristics of Soil Parameters of Agricultural Land Use Types, Their Location and Development Forecast"

_land, doi:10.3390/land9060197_

Round 1

Reviewer 1 Report

The topic is relevant for agriculture and land management. Additionally, the thematic context of the study is interesting with practical implications in (future) land uses.

There are some questions to be considered:

  1. The introduction should be improved in order to explain the necessity of this job. The objectives should be explained (line 44-48) better. I think that the Line 378 ("One objetive of this paper...") should be included in the introduction.
  2.  Conclusions - In general terms, this part is almost a summary of the results. You should improve this part to show that what you obtained (Results) is important for (future) Land management and why.

Just few thing to correct:

Line 48 : "...Geographic information systems tools (ArcGIS) tools were..."

Line 66 : 0.27 ha instead of 0,27.

Line 196 : Delete the full stop at the end of the sentence.

Line 238 :  7-12º instead of 7-12o

Line 325: Table 7: 17.4 instead of 17,4

Line 554 : 1981 (bold).

Author Response

The introduction should be improved in order to explain the necessity of this job.

Accepted.

The objectives should be explained (line 44-48) better.
Accepted.

I think that the Line 378 ("One objetive of this paper...") should be included in the introduction.
Accepted.

Conclusions - In general terms, this part is almost a summary of the results. You should improve this part to show that what you obtained (Results) is important for (future) Land management and why.
Fulfilled.

Just few thing to correct:

Line 48 : "...Geographic information systems tools (ArcGIS) tools were..."
Corrected.

Line 66 : 0.27 ha instead of 0,27.
Corrected.

Line 196 : Delete the full stop at the end of the sentence.
Corrected.

Line 238 : 7-12º instead of 7-12o
Corrected.

Line 325: Table 7: 17.4 instead of 17,4
Corrected.

Line 554 : 1981 (bold).
Corrected.

Reviewer 2 Report

General comments

The idea of the paper is of great interest. The article is based on a large amount of data and its preparation required a lot of work. The value of the work is that it presents research for the whole country. Nevertheless, the authors should make sure that the goals are clearly presented in one part of the work. I suggest that the title of the article should also include the aspect of the prognostic studies conducted.

It would also be more appropriate for the discussion to be related to the results of the research in the possible broad context, and to refer to the previous studies rather than focus on the methods used.

Specific comments:

Line 111 – translate into English

Figure 5 – you should increase the readability of the presented results.

Author Response

Nevertheless, the authors should make sure that the goals are clearly presented in one part of the work.
Added.

I suggest that the title of the article should also include the aspect of the prognostic studies conducted.
Title was changed.

It would also be more appropriate for the discussion to be related to the results of the research in the possible broad context, and to refer to the previous studies rather than focus on the methods used.
Accepted and added.

Specific comments:

Line 111 – translate into English
Deleted.

Figure 5 – you should increase the readability of the presented results.
Readability improved and corrected.

Many thanks for your article’s review – especially for professional approach and valuable advices. We have corrected our article within our capabilities and on the base of your recommendation.

This manuscript is a resubmission of an earlier submission. The following is a list of the peer review reports and author responses from that submission.